# Forced Remyelination Promotes Axon Regeneration in a Rat Model of Spinal Cord Injury

**DOI:** 10.3390/ijms24010495

**Published:** 2022-12-28

**Authors:** Małgorzata Zawadzka, Marine Yeghiazaryan, Sylwia Niedziółka, Krzysztof Miazga, Anna Kwaśniewska, Marek Bekisz, Urszula Sławińska

**Affiliations:** Nencki Institute of Experimental Biology, Polish Academy of Sciences, 3 Pasteur Str., 02-093 Warsaw, Poland

**Keywords:** demyelination/remyelination, axonal regeneration, spinal cord transection, 5-HT_2_ receptor

## Abstract

Spinal cord injuries result in the loss of motor and sensory functions controlled by neurons located at the site of the lesion and below. We hypothesized that experimentally enhanced remyelination supports axon preservation and/or growth in the total spinal cord transection in rats. Multifocal demyelination was induced by injection of ethidium bromide (EB), either at the time of transection or twice during transection and at 5 days post-injury. We demonstrated that the number of oligodendrocyte progenitor cells (OPCs) significantly increased 14 days after demyelination. Most OPCs differentiated into mature oligodendrocytes by 60–90 dpi in double-EB-injected rats; however, most axons were remyelinated by Schwann cells. A significant number of axons passed the injury epicenter and entered the distant segments of the spinal cord in the double-EB-injected rats. Moreover, some serotoninergic fibers, not detected in control animals, grew caudally through the injury site. Behavioral tests performed at 60–90 dpi revealed significant improvement in locomotor function recovery in double-EB-injected rats, which was impaired by the blockade of serotonin receptors, confirming the important role of restored serotonergic fibers in functional recovery. Our findings indicate that enhanced remyelination per se, without substantial inhibition of glial scar formation, is an important component of spinal cord injury regeneration.

## 1. Introduction

Spontaneous remyelination of residual axons often occurs after spinal cord injury, indicating an intrinsic ability of the CNS tissue to repair itself. On the other hand, the remyelination failure makes axons more sensitive to damage, deteriorates successful propagation of the action potential, and results in functional impairments. 

CNS axon remyelination is mediated by endogenous oligodendrocyte progenitor cells (OPCs) [1,2,3], which respond to demyelinating injury by undergoing activation, proliferation, migration, and differentiation into mature oligodendrocytes, restoring myelin sheaths to the denuded axons. In rodent models of experimental spinal cord injury, OPCs are efficiently recruited to the area of the lesion [4,5,6]. However, endogenous remyelination is usually not sufficient to support axonal survival and functional recovery after spinal cord injury, which was long considered to be due to the inhibitory environment created at the site of injury [7,8]. Multiple factors play a detrimental role within the injured spinal cord, wherein the differentiation of OPCs and maturation may be inhibited by a toxic environment and/or axonal ensheathment and remyelination could be affected. Actually, it has been shown that OPCs are quite effective in remyelinating nearly all intact demyelinated axons in rodents [9,10,11]. However, while efficient endogenous remyelination improves axon preservation and signal conductance [12], it is usually not capable of sufficiently supporting functional recovery after contusive spinal cord injury [9]. This is perhaps due to the delay observed in the remyelination after spinal cord injury compared to chemical demyelination [13]. Therefore, novel strategies based on enhancing endogenous remyelination in order to significant locomotor recovery should be employed early after spinal cord injury.

In contrast to traumatic injury, chemically induced focal demyelination is often followed by very fast, efficient, and extensive remyelination, which is typically complete by several weeks after toxin injection [14,15]. This regenerative process involves the local recruitment of OPCs, which then occupy demyelinated axons and differentiate into remyelinating oligodendrocytes [16]. It has been found that spontaneous remyelination of the injured spinal cord could be driven also by myelinating Schwann cells [17]. After spinal cord injury, when glia limitans are destroyed, Schwann cells of the PNS origin also easily migrate from spinal roots to the lesion, interact with demyelinated CNS axons, and myelinate them [18]. What is more, it has been shown that OPCs are also able to differentiate into Schwann-cell-like cells, producing myelin of the PNS type, as well as myelinating CNS axons and promoting their regeneration [3,17]. Several studies reported that remyelination associated with a higher proportion of Schwann cells occurs preferentially in areas where astrocytes are absent [3,19,20]. 

Remyelination, regardless of whether it is driven by oligodendrocytes or Schwann cells, results in increased expression of regeneration-associated genes in injured neurons [21] and ultimately saves them from death.

Several lines of electrophysiological evidence strongly support the notion that remyelination of chemically demyelinated axons is an important factor in reestablishing the functional conduction of demyelinated axons [22,23,24]. However, the effect of early forced remyelination on axonal outgrowth and functional regeneration following spinal cord injury has not been documented previously.

Here, we investigated the ability of forced remyelination to support axon regeneration in an acute and chronic (14 days and 2–3 months, respectively) total transection model. We used ethidium bromide (EB), a DNA intercalating agent, that kills glial cells, including astrocytes, oligodendrocytes, and OPCs, leaving the axons mostly intact. Inducing the focal demyelination within the transected spinal cord, we hypothesized that OPC distributed within normal-appearing white matter in the lesion vicinity will become activated, repopulate the lesion area, interact with demyelinated axons, and contribute to axonal regeneration and functional recovery. Our results demonstrate that the induced remyelination improves the functional recovery after transection by enhancing the oligodendrocyte precursor recruitment and differentiation, resulting in oligodendrocyte and Schwann-cell-mediated remyelination without substantial suppression of the glial scar formation. 

Considering the above, we postulate that enhancing remyelination early after traumatic insult could be permissive for axon survival and regrowth leading to the preservation of hind limb functions in a rat model of complete spinal cord transection.

## 2. Results

### 2.1. Chemically Induced White Matter Demyelination of the Transected Spinal Cord Was Followed by Subsequent Remyelination and Axonal Regrowth

Adult OPCs are able to repopulate the areas of demyelination very efficiently [25]. Moreover, it has been described that when the same area is exposed to several rounds of demyelination/remyelination, the OPC numbers are not reduced and the efficiency of remyelination is not impaired by previous rounds of remyelination [16]. Given the above, we hypothesized that de/remyelination induced close to the area of the primary traumatic lesion might beneficially modulate the lesion environment; induce more OPCs into the injured tissue; enhance subsequent remyelination; and, in consequence, support axonal survival. To test this hypothesis, we induced multifocal demyelination at approx. 1–1.5 mm both rostrally and caudally to the transection site by stereotaxic injection of the glial toxin into the dorsal and ventrolateral white matter areas through using the method described by Woodruff and Franklin [26] (Figure 1A). Adult female WAG rats were used in experimental (transection combined with demyelination induced by single or double EB injection, TTx1EB or TTx2EB, respectively) and control (only transection, TTnoEB) groups.

First, to establish whether chemically induced multifocal de/remyelination created around spinal cord traumatic lesions affects the architecture of injured tissue, we evaluated the extent of remyelination and the number of myelinated axons in animals 2–3 months post-injury (at 60–90 dpi). Toluidine blue staining of semi-thin resin cross-sections revealed the presence of extensive remyelinated areas (Figure 1B) and a significantly higher number of myelinated axons in the dorsal and ventrolateral white matter of the TTx2EB spinal cords compared to TTnoEB ones (Student’s *t*-test; *p* < 0.001; Figure 1C). In the control TTnoEB animals, total spinal cord transection resulted in severe demyelination and axonal atrophy. Remyelination was not found in the white matter, which showed mostly degraded axons and their debris within and around the transection site. Extensive axonal remyelination in the TTx2EB group was evident also in electron microscopy images. White matter axons were surrounded by myelin sheaths, mostly identified to be Schwann-cell-derived ones by their ultrastructure characteristic (Figure 1D). 

To examine the effect of forced de/remyelination on spinal cord tissue atrophy, which usually occurs after transection, we evaluated the size of intraspinal cysts created within the injured area at 60–90 dpi (Figure 1E,F). Analysis of the longitudinal sections of injured spinal cords showed a significant reduction of cyst area in cords of the TTx2EB rats compared to controls, suggesting better tissue preservation in rats with de/remyelination (Student’s *t*-test; *p* < 0.05; Figure 1E,F).

Next, we assessed the effect of forced de/remyelination on axonal growth/preservation. Axons were immunostained for neurofilaments with anti-NF-H antibody and quantified (Figure 1E and Figure 2A–C). A significantly higher number of longitudinally distributed axons at the adjacent area of the lesion were observed in the TTx2EB rats compared to the control TTnoEB animals without de/remyelination (one-way ANOVA; *p* = 0.043/F(2,11) = 4.26; post hoc test *p*-values: 0.83, 0.025, and 0.058 for TTnoEB vs. TTx1EB, TTnoEB vs. TTx2EB, and TTx1EB vs. TTx2EB, respectively). This effect was even more striking at the epicenter of the lesion where apart from the presence of a large amount of debris and fragmented axons, significantly more axons crossing the spared tissue were found in the TTx2EB animals compared to the other two experimental groups (one-way ANOVA: *p* = 0.0004/F(2,11) = 16.84; post hoc test *p*-values: 0.67, 0.0006, and 0.0006 for TTnoEB vs. TTx1EB, TTnoEB vs. TTx2EB, and TTx1EB vs. TTx2EB, respectively; Figure 2C). Interestingly, a visibly more Schwann-cell-specific protein, periaxin, was associated with myelinated axons both at the periphery of the lesions and those crossing the epicenter of the injury (Figure 2A). A substantial number of periaxin-associated, remyelinated axons were also detected within the epicenter of the lesion as late as 150 days post-injury (Figure 2B). Quantitatively, the spinal cords of the TTx2EB rats presented a significantly larger area of periaxin immunopositive fibers distributed at the adjacent area of the lesions in comparison to the control TTnoEB animals (one-way ANOVA: *p* = 0.023/F(2,6) = 7.61; post hoc test *p*-values: 0.05, 0.008, and 0.21 for TTnoEB vs. TTx1EB, TTnoEB vs. TTx2EB, and TTx1EB vs. TTx2EB, respectively). Moreover, in the epicenter of the injury, the TTx2EB rats showed a significantly larger area of periaxin immunopositive fibers than the other two experimental groups (one-way ANOVA: *p* = 0.0007/F(2,6) = 31.14; post hoc test *p*-values: 0.96, 0.0005, and 0.0005 for TTnoEB vs. TTx1EB, TTnoEB vs. TTx2EB, and TTx1EB vs. TTx2EB, respectively; Figure 2D).

Apart from quantifying the amount of peripheral myelin, we also assessed whether multifocal de/remyelination altered the number of oligodendrocyte lineage cells, the classical myelin-producing cells of the CNS, in the injured spinal cords. The identity of oligodendrocyte lineage cells and mature oligodendrocytes was confirmed by detection of Olig2^+^ and Olig2^+^/CC1^+^, respectively (Figure 3A,B). Quantification of Olig2 immunolabeled cells at 14 dpi revealed a significantly higher number of positive cells in the area adjacent to the transection in both experimental groups with the EB injection compared to the control (one-way ANOVA: *p* = 0.015/F = (2,8) = 7.51; post hoc test *p*-values: 0.005, 0.047, and 0.14 for TTnoEB vs. TTx1EB, TTnoEB vs. TTx2EB, and TTx1EB vs. TTx2EB, respectively; Figure 3C). However, in the epicenter of lesions, a markedly larger amount of Olig2-positive cells (than in the control) were present only in the TTx1EB rats (one-way ANOVA: *p* = 0.03/F(2,8) = 5.63; post hoc test *p*-values: 0.01, 0.18, and 0.08 for TTnoEB vs. TTx1EB, TTnoEB vs. TTx2EB, and TTx1EB vs. TTx2EB, respectively; Figure 3C). Although at 60–90 dpi (Figure 3D), the number of Olig2-labeled cells was detected as much lower than at 14 dpi, it remained significantly elevated compared to controls both for the area adjacent to the lesion (two-way ANOVA, treatment factor: *p* = 0.0003/F(2,19) = 12.92; post hoc test *p*-values: 0.0005, 0.02, and 0.056 for TTnoEB vs. TTx1EB, TTnoEB vs. TTx2EB, and TTx1EB vs. TTx2EB, respectively) and for the lesion epicenter (two-way ANOVA, treatment factor: *p* = 0.008/F(2,18) = 6.33; post hoc test *p*-values: 0.004, 0.011, and 0.38 for TTnoEB vs. TTx1EB, TTnoEB vs. TTx2EB, and TTx1EB vs. TTx2EB, respectively). Moreover, at 60–90 dpi, there were a greater number of Olig2/CC1-labeled mature oligodendrocytes in the area adjacent to the lesion site in the TTx2EB animals compared to the control group (Figure 3D; two-way ANOVA, treatment factor: *p* = 0.0003/F(2,19) = 12.92; post hoc test *p*-values: 0.08, 0.001, and 0.14 for TTnoEB vs. TTx1EB, TTnoEB vs. TTx2EB, and TTx1EB vs. TTx2EB, respectively). In the epicenter of the lesion, Olig2^+^/CC1^+^ cells were, however, no more numerous in the TTx2EB rats than in the controls (two-way ANOVA, treatment factor: *p* = 0.008/F(2,18) = 6.33; post hoc test *p*-values: 0.77, 0.069, and 0.17 for TTnoEB vs. TTx1EB, TTnoEB vs. TTx2EB, and TTx1EB vs. TTx2EB, respectively).

In general, at 60–90 dpi, the number of Olig2^+^/CC1^+^ cells was lower than the number of Olig2^+^ cells for both lesion-adjacent areas (two-way ANOVA, cell-type factor: *p* = 0.0004/F(1,19) = 18.13) and lesion epicenter (two-way ANOVA, cell-type factor: *p* = 0.006/F(1,18) = 9.97), but with some variation depending on the experimental group. In particular, nearly all (86%) Olig2^+^ cells in the adjacent area from the TTx2EB group were detected to also be CC1^+^ cells (*p* = 0.31, relevant post hoc test for Olig2^+^ vs. Olig2^+^/CC1^+^), contrary to controls, where such a percentage was much lower (48%; *p* = 0.057, relevant post hoc test for Olig2^+^ vs. Olig2^+^/CC1^+^). It suggests that the local microenvironment created by enhanced experimental demyelination was permissive for efficient cell differentiation into mature oligodendrocytes. 

Together, these results confirmed that de/remyelination induced by double EB injection was followed by extensive remyelination driven by both oligodendrocyte and Schwann cells and accompanied by better preservation of white matter axons.

### 2.2. Forced Demyelination and Subsequent Remyelination Moderately Altered Astro- and Microglial Response to Spinal Cord Transection

To determine whether the reactive glial response to the spinal cord transection was affected by multifocal demyelination, we counted astrocytes and microglia/macrophages activated or recruited to the lesion areas. In the spinal cords of control animals, scar tissue with strong immunoreactivity for glial fibrillary acidic protein (GFAP) was detected mostly in the border of the lesion at both 14 dpi and 60–90 dpi. We found no difference in the extent of astrogliosis between controls and EB-injected animals at 14 dpi (one-way ANOVA: *p* = 0.76/F(2,10) = 0.28—adjacent area and *p* = 0.61/F(2,10) = 0.52—epicenter) and also at 60–90 dpi for the lesion periphery (one-way ANOVA: *p* = 0.94/F(2,14) = 0.06). Activated astrocytes were evident also at the lesion epicenter at 60–90 dpi. GFAP-immunopositive staining assessed at 2–3 months post-injury showed the total area of activated astrocytes in the lesion epicenter to be noticeably smaller in the TTx2EB spinal cords than in the controls (Figure 4). However, variability of GFAP immunoreactivity in the epicenter area between different animals was rather large and, in consequence, the average value showed only a tendency to be smaller in the TTx2EB group (Figure 4C; one way ANOVA: *p* = 0.19/F(2,14) = 1.89; post hoc test for TTnoEB vs. TTx2EB: *p* = 0.08). 

Immunodetection for ionized calcium binding adaptor molecule 1 (IBA-1), which sensitively marks microglia and macrophages, revealed a large number of myeloid-derived cells occupying the injured tissue that were evident both in the adjacent scar tissue and at the epicenter of the injury in control rats at all time points (Figure 4). In the adjacent tissue, two weeks after SCI, microglia/macrophages were detected at a similar density in the spinal cord lesions in controls and the TTx1EB rats and at a slightly but significantly higher level in TTx2EB rats (one-way ANOVA: *p* = 0.003/F(2,9) = 12.1; post hoc tests: *p* = 0.15—TTnoEB vs. TTx1EB, *p* = 0.0009—TTnoEB vs. TTx2EB, *p* = 0.01—TTx1EB vs. TTx2EB). On the other hand, in the same time period at the epicenter of the injury, all experimental groups showed similar IBA-1 immunoreactivity (one-way ANOVA: *p* = 0.36/F(2,9) = 1.14). Two to three months after SCI, the total IBA-1 immunoreactive area was found to be significantly lower in the TTx2EB rats for either the adjacent scar tissue (Figure 4D; one-way ANOVA: *p* = 0.016/F(2,13) = 5.81; post-hoc test *p*-values: 0.048. 0.005, and 0.65 for TTnoEB vs. TTx1B, TTnoEB vs. TTx2EB, and TTx1EB vs. TTx2EB, respectively) or the lesion epicenter (Figure 4D; one-way Brown–Forsythe ANOVA: *p* = 0.03/F(2,8.3) = 5.47; post hoc test *p*-values: 0.52. 0.041, and 0.055 for TTnoEB vs. TTx1B, TTnoEB vs. TTx2EB, and TTx1EB vs. TTx2EB, respectively). 

Histological analyses of gliosis therefore revealed a quite moderate effect of demyelination on cell proliferation and migration to the different lesion parts; however, it seemed that the loss of astrocytes caused by EB injection within the epicenter area of the lesion was stronger than in the transected tissue, which could be causative for Schwann cell invasion and/or differentiation. 

### 2.3. Locomotor Performance Was Improved in Rats with Forced Remyelination

Since we observed the most significant histological effects of forced remyelination in the TTx2EB rats, we then tested the locomotor hindlimb movement performance of these rats and TTnoEB controls at 2–3 months after spinal cord transection. For the locomotor hindlimb movement testing, the rats were placed with their forelimbs and forequarters on a platform above the treadmill while the hindlimbs were touching the moving belt. In such experimental conditions, a tail pinch triggers hindlimb movements. In the TTx2EB rats, tail pinching first induced the hindlimb extension, promoting a high body posture (Figure 5A top) that was followed by episodes of consistent hindlimb plantar stepping lasting for several seconds. Although spontaneous locomotor hindlimb movement abilities in an open field did not show any improvement, the functional hindlimb locomotor movement induced by tail pinching on a treadmill revealed significant differences between EB-treated and untreated animals. The episodes of plantar stepping in the TTx2EB rats induced by tail pinching were associated with rhythmic alternating EMG burst activity of the Sol and TA muscles (Figure 5B top), with the remaining muscle activity obtained during proper plantar walking of the intact rats (see Figure 1 in [27]). In contrast to the TTx2EB rats, in the control TTnoEB rats, the tail pinching was able to induce only limited uncoordinated movements of both hindlimbs with the feet dragging on the dorsal surface over the moving belt (Figure 5A bottom), which was accompanied by uncoordinated irregular EMG burst activity of Sol and TA muscles (see Figure 5B bottom). 

Next, we used the EMG activity recordings to perform a precise analysis of locomotor pattern correctness. On the basis of the EMG activity of Sol (ankle extensor) and TA (ankle flexor) muscles of both hindlimbs recorded in the TTx2EB and the untreated TTnoEB (control) animals, we compared the quality of locomotor performance by analyzing the step cycle duration (Figure 5E) as well as the EMG of burst duration (Figure 5F,G) and peak amplitude (Figure 5H,I). In addition, the inter- and intralimb coordination of hindlimb movements was analyzed (Figure 5C,D). 

Both interlimb and intralimb coordination became better in the TTx2EB rats with local demyelination as compared to the control TTnoEB animals (Figure 5C,D). The strength (***r***-vector length) of the interlimb coordination (left vs. right soleus) was higher in the TTx2EB rats than in the TTnoEB rats (*p* = 0.01, Mann-Whitney), and the strength of the intralimb coordination (Sol vs. TA muscles) was higher in the TTx2EB than in TTnoEB rats (*p* = 0.03, Mann-Whitney). These indicate that the locomotor pattern of the rats with induced local remyelination was coordinated significantly better than that of the control animals without EB treatment.

Additionally, the phase shift of interlimb coordination measured for the Sol EMG burst onsets showed significant differences between remyelinated and control rats (*p* = 0.03, Watson U^2^ test), indicating more symmetrical alternating (closer to 180 deg) activity of the left and right Sol muscles in the TTx2EB rats. No significant difference in phase shift between Sol and TA muscles in the same limb (*p* = 0.34, Mann-Whitney) was observed.

Next, we noted that the step cycle duration established on the basis of the Sol burst activity did not differ between the control TTnoEB and the treated TTx2EB rats (*p* = 0.96, Student’s *t*-test) (Figure 5E). However, the relative variability of this parameter measured for individual rats as a coefficient of variation (CV = SD/mean) was significantly smaller in the TTx2EB rats than in the TTnoEB controls (*p* = 0.013, Student’s *t*-test). A lower coefficient of variation means that the locomotion cycle in the TTx2EB rats was more stable and consistent than in the untreated TTnoEB control rats. 

In the TTx2EB rats, the Sol burst duration (Figure 5F) was not different from that of TTnoEB rats (*p* = 0.22, Student’s *t*-test). Moreover, the relative variability of Sol burst duration was not different in both groups (for respective CV value comparison *p* = 0.23, Student’s *t*-test). On the other hand, the TA muscle in the TTx2EB rats had a much shorter EMG burst duration (Figure 5G) than that in the TTnoEB control animals (*p* = 0.0008, Student’s *t*-test) and was associated with smaller relative variability revealed by the mean CV values (*p* = 0.0015, Student’s *t*-test). Thus, our EMG analysis indicated that in the TTx2EB rats, the swing phase (the foot dorsiflexion related to the TA EMG burst activity) was not only shorter but more stable as well as consistent than that in control TTnoEB animals. This is consistent with our observation that the treated rats presented hindlimb plantar stepping associated with well-pronounced foot dorsiflexion and toe clearing at the swing phase during episodes of tail-pinching-induced locomotor performance.

Next, we analyzed the EMG burst amplitude (EMG amplitude was determined from filtered (bandpass 0.1–1 kHz), integrated (20 Hz), and rectified EMG records of at least 10 consecutive steps). In the Sol muscle, the amplitude of EMG bursts (Figure 5H) was substantially higher in the TTx2EB rats (by 83%) as compared to the control TTnoEB animals (*p* = 0.032, Student’s *t*-test). Moreover, the EMG burst amplitude variability measured as CV values obtained during the locomotion of individual rats from the TTx2EB group was 31% lower than in the control group (*p* = 0.021, Student’s *t*-test). For the TA muscle, we did not notice any tendency for higher EMG amplitude (Figure 5I) in the TTx2EB rats (*p* = 0.78, Student’s *t*-test). However, similar to Sol, the TA muscle amplitude variability (CV values) was lower for the TTx2EB rats than that for control TTnoEB animals (*p* = 0.05, Student’s *t*-test).

Thus, our results indicate that the soleus muscle activity was stronger and the strength of both muscle activities was more stable and consistent in the TTx2EB rats than in the control TTnoEB animals.

### 2.4. The Blockade of 5-HT_2_ Receptors Affected the Locomotor Performance of the Hind Limbs

It has been previously reported that the presence of serotonergic fibers caudal to a complete spinal cord transection is associated with locomotor function improvement [28,29]. Therefore, we examined whether the serotonergic fibers were present in the injured spinal cord of EB-treated rats and, if so, what their role was in observed functional improvement. First, we immunostained the spinal cord cross- and longitudinal sections with anti-5-HT antibody and found, as expected, a normal pattern of numerous serotonin-positive axons in the spinal cords of both the TTnoEB (Figure 6A, left image) and TTx2EB (not shown) rats above the transection site. Interestingly, 5-HT-positive axons were detected caudally, below the transection site, in the rats from the TTx2EB group and occasionally in the rats from the TTx1EB group, in contrast to controls that were completely devoid of serotonergic innervation (Figure 6A; middle image). Moreover, serotonergic axons in the TTx2EB rats finally reached the L2-L5 level of the spinal cord below the total transection site (Figure 6A; right image). In the transverse sections of the TTx2EB spinal cords, we observed serotonergic axons extended through the scar and entered the lesion epicenter (Figure 6B; middle and right image), which was not seen in the control TTnoEB rats (Figure 6B; left image). 

Next, we sought to understand the role of the serotonergic innervation in the TTx2EB rats. To this end, we tested the effects of 5-HT_2_ receptor antagonist (cyproheptadine; 1 mg/kg i.p.) on locomotor hindlimb movement induced by tail pinching. Blockade of the 5-HT_2_ receptors by cyproheptadine induced a significant alteration of hindlimb locomotor performance. In the pre-drug conditions, the spinal rats from the TTx2EB group could be induced by tail pinching to present a distinctive locomotor posture that was followed by several nice plantar stepping performances accompanied by the rhythmic EMG burst activity of hindlimb Sol and TA muscles (top in Figure 7A,B). During 5 to 15 min after cyproheptadine application (i.e., when the first effect of the drug administration was usually observed), the rats lost the ability to perform any hindlimb plantar stepping and the reduced posture was followed by limb dragging movement accompanied by irregular EMG activity with low amplitude (bottom in Figure 7A,B). After cyproheptadine treatment, inter- and intralimb coordination of hindlimb movements was much more unstable and irregular. The phases of interlimb coordination (between the activity of left and right Sol EMG burst activity, Figure 7C) as well as intralimb coordination (between Soleus and TA EMG burst activity in the same limb, Figure 7D) were for individual steps much more uniformly distributed within the full 360 degrees. In the post-drug condition, the phase of interlimb coordination (angle of ***r***-vector in the polar plot), as well as the intralimb coordination, did not differ from those observed in the control pre-drug period (*p* = 0.65 and *p* = 0.27, Moore’s test for paired angular data). However, due to irregularity, the inter- and intralimb coordination strength (i.e., length of ***r***-vector) substantially decreased by 66.6 % and 37.8% (respectively, *p* = 0.0007 and *p* = 0.0037, one-sample *t*-test) as compared to the control situation (Figure 7E,F; compare also red ***r***-vector lengths in Figure 7C,D). Cyproheptadine did not change cycle duration (Figure 7G; *p* = 0.87, one-sample *t*-test) but greatly influenced the duration and amplitude of EMG bursts. For the Soleus muscle, the EMG burst duration and amplitude decreased by 34.9% and 55.9% (respectively, *p* < 0.0001 and *p* = 0.0002, one-sample *t*-test) as compared to the respective values from the control pre-drug condition (Figure 7H,I). For the TA muscle, EMG burst duration became more than two times longer (*p* = 0.0045, one-sample *t-*test), but EMG burst amplitude was almost two times smaller (*p* = 0.0006, one-sample *t*-test) than in the control state.

Thus, our results indicate that 5-HT fibers passing through and elongated caudally to the lesion site are responsible, at least partially, for the hindlimb locomotor recovery observed in the TTx2EB rats.

## 3. Discussion

Spinal cord injury (SCI) is a devastating neurological condition leading to severe dysfunctions of sensory, motor, and autonomic functions. Deficient remyelination of affected axons is considered one of the major pathological factors that contribute to poor functional recovery [30,31] since the demyelinated axons are vulnerable to damage in the lesion microenvironment that ultimately leads to axonal degeneration [32]. However, even though endogenous OPCs present in the lesion vicinity have the potential to differentiate into mature oligodendrocytes serving the injured axons with new supportive myelin sheaths, remyelination in spinal cord injuries often fails or remains incomplete. Poor recovery after SCI may be, therefore, largely due to the limited self-renewal capacity of oligodendrocyte progenitors or their limited differentiation. Previous studies have reported some promising results on the repopulation of OPCs by their transplantation to promote myelination [33,34]. Nevertheless, the rescuing effects are often compromised by the hostile microenvironment of the injury site, resulting in inadequate remyelination (for a review, see [35]). Thus, we hypothesized that the experimentally enhanced endogenous remyelination could be a promising target for axonal protection after SCI and, ultimately, functional recovery.

The studies reported here demonstrate that the forced white matter remyelination promotes axonal regeneration in a rat model of spinal cord injury. Our findings indicate that remyelination induced within the normal appeared white matter around the transection site significantly enhanced OPC recruitment and differentiation, Schwann cell invasion/differentiation, and spontaneous axon growth throughout the transection site. Finally, we show that enhanced remyelination supports the regeneration of serotoninergic descending fibers which, at least in part, promotes functional recovery. 

In this study, we combined two well-established spinal cord injury models: a complete transection of the spinal cord followed by the stereotactic injection of a nucleic acid chelator, ethidium bromide, that induces demyelination to the surrounding white matter. This elaborate procedure creates, to our best knowledge, the most unbiased approach for exploring the role of remyelination in tissue reconstruction after spinal cord injury. First, total transection leaves the spinal cord with virtually no spared axons at the defined lesion site, which eliminates spontaneous regenerative mechanisms that occur in partial injuries, the effects of which may mask the immediate results of the applied strategy in an uncontrolled way. Next, the toxin that is injected into the normal-appearing white matter within the primary lesion vicinity kills glial cells (mainly oligodendrocytes, astrocytes, and OPCs), ultimately causing primary focal demyelination early after injection, but saves the axons within the affected area intact [6,26]. Finally, particularly important is the fact that the neural structures located below the injury remain functionally active, and therefore it is possible to identify mechanisms (including endogenous ones) that might be activated to improve motor functions [27,36,37,38,39,40,41,42]. All of the above makes this model simple to standardize, clear, reliable, and highly reproducible and thus useful for pharmacological or cellular studies focused on axonal regeneration.

Here, we demonstrate that at 14 dpi after SCI followed by EB injection, the number of OPCs at the adjacent area of transection was greater than those in control animals. It has been already clearly proved that oligodendrocyte progenitor cells migrate from the surrounding intact white matter into the demyelinated area [43], proliferate repopulating the lesion area, and finally differentiate into mature oligodendrocytes responsible for remyelination [3,44,45]. We observed a profound increase in OPCs and mature oligodendrocytes at the epicenter of the SCI in the EB-injected rats compared to the control animals. Moreover, in the selected experimental group (TTx2EB), we repeated EB injection 5 days after the initial one, hypothesizing that it would mobilize significantly more OPCs that could be engaged with affected axons and serve them with myelin sheaths. Indeed, we found that the number of OPCs increased sharply between TTx1EB and TTx2EB groups at 14 dpi, together with the increased proportion of CC1^+^ cells among Olig2^+^ cells. 

The success of OPC recruitment and migration after demyelination is limited by several critical factors including inhibitory glial scarring formation. Glial cells that repopulate lesion areas secrete molecules that directly inhibit OPC maturation and subsequent remyelination. Moreover, the presence of myelin-associated debris within the lesion, including myelin-associated glycoprotein (MAG), myelin-associated glycoprotein (MOG), and Nogo-A, inhibits OPCs differentiation and prevents axonal regeneration [7,46]. In our study, we found that after EB-induced demyelination, the number of astrocytes was not altered at 14 dpi in both the lesion epicenter and adjacent area compared to the transected cords. At 60–90 dpi, almost no astrocytes existed in the lesion epicenter of double-injected cords since they were likely killed by ethidium bromide. However, we detected that astrocytes repopulated the transection adjacent area very efficiently, and reactive astrogliosis was significantly higher compared to 14 dpi, regardless of the experimental condition. 

The response and infiltration of IBA-1^+^ inflammatory cells appeared relatively equal regarding cell morphology, number, and distribution at early remyelination (14 dpi) but decreased with time, and at 60–90 dpi was weakest in double-injected cords. Our results show that the significance of postlesion gliosis early after injury may be less than assumed since the glial scar created in response to the total transection was generally not altered by the demyelinating factor or the changes turned out to be relatively small, contrary to expectations. 

Our data imply that enhancement of remyelination per se might be one of the critical factors for the regeneration of spinal cord axons and exhaustion of OPCs in the demyelinated area or their defective differentiation into new oligodendrocytes might be one of the reasons for its failure. Since endogenous oligodendrocyte remyelination of spared axons is not efficient enough to restore spontaneous locomotor recovery following contusive SCI [4], two strategies could be proposed to overcome this obstacle: cell transplantation or acceleration of endogenous remyelination. 

It is generally accepted that increased remyelination following transplantation of cells that are able to differentiate into oligodendrocytes correlates with improved functional recovery in animal models of spinal cord injury [34,47,48,49,50]. However, this may not always be the case since, for example, transplantation of mouse platelet-derived growth factor (PDGF)-responsive neural precursors after a rat contusion injury neither improved the remyelination rate nor provided functional benefit [51]. This suggests that the injury-created microenvironment may control remyelination and its modulation could be an attractive target for improving axonal sparing. On the other hand, it is likely that myelin-derived protection of affected axons could be one of the possible mechanisms underlying the correlation between remyelination and functional recovery after SCI [8]. Our results suggested that the presence of excess OPCs in the lesion vicinity, at the early phase of injury, may also provide important support to the integrity of axons. 

Cell transplantation studies provide some evidence in support of such a notion: transplanted OPCs could, apart from producing myelinated oligodendrocytes, secrete trophic factors [52], contribute to the glial scar and regulate axonal growth [53,54], modulate inflammatory responses [55,56], or promote endogenous remyelination [57]. 

In our study, we demonstrated that at 2–3 months following spinal cord injury, both the recruited CNS cells, as well as the invading PNS-derived ones, repopulated and efficiently remyelinated CNS axons. It was shown that Schwann cells were able to migrate into the CNS, but their survival was limited and the presence of peripheral myelin was restricted to the spinal root’s entry and exit zones (for a review, see [58]) and close to blood vessels [18]. However, Schwann cells can be frequently found within the CNS in pathological conditions such as demyelinating lesions. These cells migrate either from the PNS [59] or they might be derived from oligodendrocyte progenitors. Genetic fate mapping has shown that the majority of these Schwann cells are derived from OPCs [3,17]. Moreover, extensive remyelination mediated by CNS-derived Schwann cells was confirmed following clinically relevant traumatic spinal contusion injury [13]. Although mechanisms underlying the Schwann cell fate of CNS progenitors remain elusive, it is well documented that a lack of astrocytes in the affected area is necessary for such a transition [20,60,61]. Nevertheless, compelling evidence indicates that regardless of their origin, Schwann cells serve CNS axons with peripheral myelin that protects their structure; restores axonal conduction; and, in some cases, reverses neurological deficits [24,62,63,64]. As learned from cell transplantation studies, the presence of Schwann cells in the CNS lesions may have several beneficial effects apart from remyelination itself, such as bridging the lesion cavity and enhancing axonal growth and sprouting through the lesion site. It has been proposed that Schwann cells transplanted into the transected spinal cord enable elongation of astrocyte processes and formation of glial-derived tunnels that support brainstem axon regeneration. Brainstem axons that regenerate across the Schwann cell bridge mediate improvement in hindlimb movement through extrasynaptic release of neurotransmitter and/or the potential formation of synapses at the caudal bridge–spinal cord interface [65]. Furthermore, skin-precursor-derived Schwann cells transplanted into the contused rat spinal cord as late as eight weeks post-injury were shown to mitigate glial scar formation, form cellular conduits to bridge and filled much of the lesions, enhance the presence of endogenous Schwann cells, and myelinate host axons, ultimately significantly improving locomotor outcomes [66].

In this study, we report that experimentally induced, endogenous white matter remyelination supported axon growth in the transected spinal cord and improvement locomotion. Moreover, we show that this beneficial effect was mediated, at least partially, by the preservation or regeneration of serotonergic descending axons across the site of total transection. Functionally, a blockade of 5-HT_2_ receptors induced a significant alteration of locomotor performance in the TTx2EB rats. This suggests that the regenerated serotonergic descending axons in the rat transected spinal cord with induced remyelination were able to mediate a functional serotonergic innervation responsible for significant hindlimb locomotion recovery. A similar effect of locomotor movement alteration was obtained by the blockade of 5-HT_2_ receptors in the lumbar spinal cord of adult intact rats [67]. Moreover, in adult rats with spinal cord total transection, the recovery of hindlimb locomotor movement can be enhanced by the activation of 5-HT_2_ serotonergic receptors [42]. A strong correlation between recovery of locomotor function and the presence of 5-HT fibers and terminals in the lumbar cord was also demonstrated [68,69,70]. Moreover, the intraspinal grafting of brainstem embryonic tissue containing serotonergic neurons enhanced the locomotor recovery of hindlimb movements, being 5-HT receptor dependent [29,71]. The observed serotonin-related locomotor recovery was consistent with the well-known crucial role of serotonin in the control of locomotor behavior (for a review, see [72,73,74,75,76]).

There is evidence emphasizing that serotonergic fibers survive CNS injury and sprout better than other neuronal subtypes [77,78,79]. The comparison of the regenerative capacity of cortical versus serotonergic fibers (brainstem origin) revealed that serotonergic neurons maintain a more active growth cone and have increased GAP-43 and increased β1 integrin subunit expression [77,80], which can be partially responsible for the robust behavior of serotonergic neurons in the injured CNS environment. Moreover, serotonergic neurons die less due to macrophage activity and respond more vigorously to various forms of experimental modulation [77,81,82]. After intraspinal grafting into the injured adult spinal cord, serotonergic neurons of embryonal origin can extend axons and incorporate remarkably well into the host spinal cord and enhance locomotor recovery [29,83,84,85]. Further experiments are needed to identify mechanisms underlying the role of remyelination in this phenomenon. 

In summary, our results show that experimentally enhanced, endogenous remyelination per se could be considered an important part of recovery after spinal cord injury. Therapeutical strategies to mobilize endogenous OPCs and/or Schwann cells to migrate into the injury site and enhance remyelination of the affected axons may substantially support their survival and ultimately locomotor recovery. Recent advances in understanding the complex mechanisms of remyelination and the well-documented potential of its modulation achieved either by pharmacological promoting of endogenous OPC migration and subsequent differentiation [86,87] or modulating the inflammatory reaction [88,89] could be also employed in spinal cord injury treatment.

## 4. Materials and Methods

### 4.1. Animal Model and Ethics Statement

Wistar Albino Glaxo (WAG) female rats, 3 months old at the start of the experiment, were used in this study. All animals were housed in the same facility with a 12/12 h light-dark cycle in the animal house of the Nencki Institute of Experimental Biology. The animal procedures were performed according to the guidelines from Directive 2010/63/EU of the European Parliament on the protection of animals used for scientific purposes and were approved by the First Ethical Committee on Animal Experiments in Poland (decision no. 256/2012).

All efforts were made to minimize animal suffering and decrease the number of animals used. During all surgical procedures, animals were anesthetized with isoflurane supplemented with Butomidor (butorphanol, s.c., 0.05 mg/kg b.w.) for pain relief.

The animals were randomly assigned to three experimental groups. In the control group, rats received total spinal cord transection using the procedure described below with no additional pharmacological treatment (TTnoEB). The second group of rats was injected once with glial toxin ethidium bromide (EB) immediately after spinal cord transection (TTx1EB). A third group also received a second injection of EB on day 5 after the initial transection and the first EB injection (TTx2EB). The animals were sacrificed 14 days after spinal cord transection for initial histology or 2–3 months after spinal cord transection for final histology. Two to three months after transection, the functional EMG recordings were also carried out. 

### 4.2. Complete Spinal Cord Transection

The spinal cord transection was performed under deep isoflurane anesthesia (Isofluorane, 2% and Butomidor, 0.05 mg/kg b.w.) using the modified surgical procedures described in detail in our previous papers (e.g., [29]). Briefly, at first, the back of the animal was shaved and disinfected using 70% alcohol washes. Next, after performing a skin incision over the Th8–12 spinal cord vertebrae and the trunk muscle separation from the spine, a dorsal laminectomy was performed. Then, the dura was opened, and the spinal cord was transected using iridectomy scissors at the thoracic level Th9/10. The spinal cord was inspected under a surgical microscope to ensure complete transection. Then, the muscles and fascia overlying the paravertebral muscles were closed in layers using sterile sutures, and the skin was closed with stainless-steel surgical clips. 

After spinal cord surgery, the animals received a non-steroidal anti-inflammatory and analgesic treatment (s.c., Tolfedine 4 mg/kg b.w.) and were given antibiotics (s.c., Baytril 5 mg/kg b.w., Gentamicin 2 mg/kg b.w.) for the following 5 days. The bladder was emptied manually twice a day until the voiding reflex was re-established (about 7 days). 

### 4.3. Induction of Multifocal Demyelination 

Demyelination was induced by stereotaxic injection of 1 µL of 0.1% ethidium bromide (EB, Sigma-Aldrich, Steinheim, Germany) solution through a glass-tipped Hamilton syringe into the dorsal and ventrolateral white matter funiculi above and below the transection at a distance of 1–1.5 mm, as previously described in detail [26]. Briefly, the position of total transection was identified, exposed, and carefully cleared; the central vein was identified; and the dura was pierced with a dental needle lateral to the vein. A three-way manipulator was then used to position the needle for stereotaxic injection of EB. A Hamilton needle with a fine glass tip was advanced through the pierced dura at an angle appropriate for ventrolateral or dorsal funiculus injection. The injection was controlled at 1 µL per minute, and the needle remained in the injection site for 2 min to allow maximal diffusion of the toxin. For the second injection of EB, 5 days after the initial transection, the wound was reopened and an injection of EB was performed.

### 4.4. Implantation of EMG Recording Electrodes

Two to three months after the spinal cord total transection, the bipolar electrodes for EMG (electromyographic) recording were implanted in the soleus (Sol) and tibialis anterior (TA) muscles of both hindlimbs in both the TTnoEB (*n* = 7) and TTx2EB rats (*n* = 5) under isoflurane anesthesia (5% induction, then 2% in oxygen 0.2–0.3 L/min), as previously described [29,71,85]. The EMG electrodes were made of Teflon-coated stainless-steel wire (0.24 mm in diameter; AS633, CoonerWire Co., Chatsworth, CA, USA). One end of the electrodes was secured to the connector, covered with dental cement (Spofa Dental, Prague, Czechia) and silicone (3140 RTV, Dow Corning, Midland, MI, USA), which was secured under the skin to the back muscles of the animal. The other ends of electrodes with the hook tips with 1–1.5 mm of the insulation removed were pulled through a cutaneous incision on the back of the animal towards the selected hindlimb muscles to which they were secured by a suture after insertion with a 1–2 mm distance between the electrode tips. The ground electrode was secured under the skin on the back of the animal at some distance from the hindlimb muscles. 

### 4.5. Video and Electromyographic Recordings

One week after implantation of the EMG electrodes, the hindlimb locomotor ability was investigated during hindlimb movements induced by tail pinching in rats placed on a treadmill (Panlab, Barcelona, Spain/Harvard Apparatus, Cambridge, MA, USA) and the hindlimbs touching the moving belt (5 cm/s) with their forequarter secured on a platform above the belt. EMG activity during locomotor-like hindlimb movements of the spinal rats with and without EB treatment was recorded simultaneously with synchronized video recordings. The EMG signal was filtered (0.1 to 1 kHz bandpass), digitized (2 kHz sampling frequency), and stored on a computer using the Winnipeg Spinal Cord Research Centre data capture and analysis system (http://www.scrc.umanitoba.ca/doc/, accessed on 28 September 2022). 

### 4.6. Evaluation of Hindlimb Locomotion Based on EMG Analysis

The locomotor hindlimb movement was evaluated on the basis of analysis of the EMG activity recorded from hindlimb muscles in rats placed on the treadmill with hindlimbs touching the moving belt while their forelimbs and forequarters were kept on a platform above the belt. In such an experimental condition, a tail pinch was used to elicit hindlimb movements. The effects of tail pinching were evaluated considering the possibility of inducing hindlimb plantar stepping that can be characterized by a high body posture and the hindlimb paws touching the ground convenient to initiate locomotor stepping. Then, the continued tail pinching was tested considering the induction of hindlimbs to carry out coordinated plantar stepping with the high body posture and pelvis lifted off the ground. The plantar stepping was evaluated, considering the consistent interlimb movement associated with alternating EMG burst activity of the left and right as well as of the flexor and extensor EMG burst activity of homonymous muscles of both hindlimbs. During the sustained locomotor hindlimb performance, the EMG burst of soleus activity is related to the stance phase of the step cycle, while a brief TA EMG burst activity is related to the swing phase. 

After blinding the identity of each rat group to the examiners, the locomotor EMG pattern was analyzed using custom software (http://www.scrc.umanitoba.ca/doc/, accessed on 28 September 2022). First, the raw EMG was rectified and integrated with a 5 ms interval. Next, the episodes of the consistent EMG burst (at least 10 bursts of consecutive rhythmic activity or no less than 10 s of irregular activity) activity were identified. On the basis of the marked burst onsets and offsets, the cycle duration (time between two consecutive EMG bursts) and burst duration (time between the onset and offset of the EMG burst) in all investigated rats were established. To determine whether the inter- and intralimb coordination ***r***-values of polar plot analysis were established using Rayleigh’s circular statistical test. In the polar plots, the phase position of the ***r***-vector at 0 or 360° reflects the synchrony of analyzed EMG burst onsets, whereas 180° is equivalent to alternation. The length of the ***r***-vector (***r***-value ranging from 0 to 1) specifies the strength of coordination between analyzed muscle burst onsets. In our analysis, Rayleigh’s circular statistical test was applied to determine whether the interlimb and intralimb coordination ***r***-values were concentrated, suggesting coupling of burst activity, or dispersed, indicating no burst coordination. We considered the interlimb and intralimb coordination to be phase related when the ***r***-value was greater than the critical Rayleigh’s value (cR) for a given *p*-value [90]. For a more detailed description, see our previous publications [28,67].

### 4.7. Blockade of 5-HT_2_ Receptors by Cyproheptadine Administration

After completing the EMG data recording of locomotor-like hindlimb movements in rats with local remyelination from the TTx2EB group, the effect of a blockade of 5-HT_2_ receptors using intraperitoneal (i.p.) administration of cyproheptadine (the 5-HT_2_ antagonist) for hindlimb locomotor performance was evaluated. Cyproheptadine (i.p. 1 mg/kg b.w.; Sigma) was dissolved first in a drop of 10% of propylene glycol, then in saline. Before drug application, the pre-drug baseline performance was established on the treadmill with the 5 cm/s belt speed. The effects of 5-HT_2_ antagonist on locomotor performance were tested 10–15 min after drug injection (i.e., when the clear alteration of the locomotor ability before the total hindlimb paralysis was usually observed for at least 4–5 h). For EMG analysis, we selected the episodes of consistent EMG burst activity (at least 10 bursts of consecutive rhythmic EMG activity or no less than 10 s of irregular activity). 

### 4.8. Statistical Analysis of the Locomotor Performance Based on the EMG Data

Since the relevant datasets did not deviate from the normal distribution (D’Agostino–Pearson normality test), the significance of differences between the raw cycle duration, EMG burst duration, and EMG burst amplitude values obtained for the control (TTnoEB) and treated (TTx2EB) groups of rats was checked using unpaired Student’s *t*-tests. For the same reason, a one-sample *t*-test (for the difference from 1) was applied for all post-cyproheptadine data expressed as ratios to control pre-drug values. Since the raw lengths (***r***-values) of the ***r***-vector ranged from 0 to 1, for EMG onset, burst coordination strength comparison between the control (TTnoEB) and treated (TTx2EB) group non-parametric Mann-Whitney test was used. EMG burst phase shift comparison was carried out using the method specifically created for two-sample angular/circular data unpaired Watson U^2^ or paired Moore’s tests. The inter-animal variability of some of the analyzed parameters was described and compared using a coefficient of variation (CV = standard deviation (SD)/mean). Statistical calculations were performed in GraphPad Prism (GraphPad Software LLC, San Diego, CA, USA) except for the Watson U^2^ and Moore’s tests, which were executed in Igor Pro (WaveMetrics Inc., Portland, OR, USA). The minimum significance (two-tailed) was set at *p* = 0.05. All values of EMG parameters are reported as mean ± SEM.

### 4.9. Tissue Processing

For immunodetection, animals were terminally anesthetized with pentobarbitone and intracardially perfused with 4% (*w*/*v*) paraformaldehyde (PFA, Sigma) in phosphate-buffered saline (PBS, pH 7.4) at the indicated time after the surgical procedure. Histological analysis of the spinal cords of animals from three studied groups was performed at 14 days (control TTnoEB, TTx1EB, and TTx2EB groups) or 2–3 months (control TTnoEB, TTx1EB, and TTx2EB groups) after surgical procedures. In some cases of the TTx2EB rats, the histological analysis was performed as late as 150 dpi (days post-injury).

Lesion-contained tissue was dissected, post-fixed in 4% PFA overnight, then cryoprotected gradually in up to 30% sucrose solution prepared with PBS before embedding with an optimal cutting temperature compound (OCT embedding matrix, CellPath, Newtown, UK). Cross- and longitudinal 12 μm cryosections were thaw-mounted onto poly-L-lysine coated slides and stored at −80 °C until further use. Sections were blocked with 10% donkey serum in PBS for 2 h at room temperature and incubated for 16 h at 4 °C with primary antibodies. Then, sections were incubated with appropriate Alexa Fluor: 488-, 555-, or 647- conjugated secondary antibodies (1:500, ThermoFisher Scientific, Pittsburg, PA, USA) for 2 h at RT. The following primary antibodies were used: IBA-1 (rabbit, 1:1000, Wako, Osaka, Japan), Olig2 (rabbit, 1:500, Millipore, Temecula, CA, USA), GFAP (rabbit, 1:1000, Dako, Agilent, Santa Clara, CA, USA), CC1 (mouse, 1:200, Calbiochem, SanDiego, CA, USA), neurofilament (mouse, NF, 1:100, Dako, Agilent, Santa Clara, CA, USA), 5-HT (goat, 1:100, Immunostar, Hudson, WI, USA), and periaxin (rabbit, 1:3000, a gift from prof. Peter Brophy, Centre for Neuroscience Research, University of Edinburgh, UK). Nuclei were visualized with Topro dye (ThermoFisher Scientific, Pittsburg, PA, USA) diluted at 1:500 in PBS for 10 min at room temperature (RT). Images were examined with Zeiss Fluorescence Microscope or Leica SP5 Laser Scanning Confocal Microscope. Images were analyzed and staining intensity was quantified with ImageJ software (https://imagej.nih.gov, accessed on 21 October 2022). 

A defined, lesion-containing region (identified as an area with a higher-than-normal density of Topro stained nuclei) was selected, and images from the injury epicenter, rostral, and caudal segments adjacent to the lesion area were acquired. The quantitative assessment of the areas immunoreactive for GFAP, IBA-1, neurofilament, and periaxin in all experimental groups was performed using ImageJ software. Briefly, color immunostained images were converted to 8-bit grayscale and subjected to threshold processing, and the total area of positive staining above the threshold was calculated. The total number of Olig2- and CC1-positive cells with visible nuclei within the observed area of each section was counted. 

### 4.10. Semi-Thin Resin Sections, Rank Analysis, and Electron Microscopy Analysis

Some animals were perfused with 4% glutaraldehyde for electron microscopy. Tissue was post-fixed in glutaraldehyde solution and then cut transversely into 1.0 mm thick blocks. Blocks were further fixed in osmium tetroxide, dehydrated through ascending ethanol washes, and embedded in TAAB resin. 

Semi-thin resin sections (0.5–1.0 μm) of the toluidine-blue-stained spinal cords were analyzed to assess normal myelin, demyelinated axons, and extent of remyelination. Remyelinated axons were distinguished from normally myelinated axons outside of the lesion by the thinness of the myelin sheath. Within the lesion, demyelinated axons were identified by the absence of a myelin sheath, while remyelinated axons possess myelin sheaths with a dark staining rim and that are thinner than would be expected for the axonal diameter. Photographs were taken under a Leica microscope (Leica, DM4000B) and digital camera. Using these morphological criteria, the number of remyelinated axons was counted under blind conditions. For ultrastructural analysis, selected blocks of resin-embedded tissue were trimmed, and ultrathin sections (50–70 nm) were cut onto copper grids. Sections were stained with lead citrate and uranyl acetate according to standard protocols before being examined by transmission electron microscope—JEM 1400 (JEOL Co., Tokyo, Japan).

### 4.11. Statistical Analysis of Immunohistochemical Data

To detect differences between immunohistochemical parameters from three different experimental groups (TTnoEB, TTx1EB, and TTx2EB), ordinary one-way ANOVA was applied. Ordinary two-way ANOVA was used to compare data from the three experimental conditions and, at the same time, from two types of cell immunostaining (using Olig2 and Olig2/CC-1 antibodies). In all cases, ANOVA analysis was calculated separately for the area adjacent to the place of injury caused by TT and for the injury epicenter and was followed by post hoc comparisons applying Fisher’s LSD tests. In one case when there were significant differences between standard deviations of data obtained for different experimental groups, the one-way Brown–Forsythe ANOVA was applied with post hoc Welch-corrected Student’s *t*-tests. Two histological parameters were compared only between the TTnoEB and TTx2EB groups, and in such a situation, standard Student’s *t*-tests were used to test differences. The entire statistical analysis of histological data was performed in GraphPad Prism (GraphPad Software LLC, San Diego, CA, USA). All values of parameters obtained from the histological analysis are reported as mean ± SEM. For all tests, *p* = 0.05 was taken as the minimum level of statistical significance. 

## Figures and Tables

**Figure 1 ijms-24-00495-f001:**
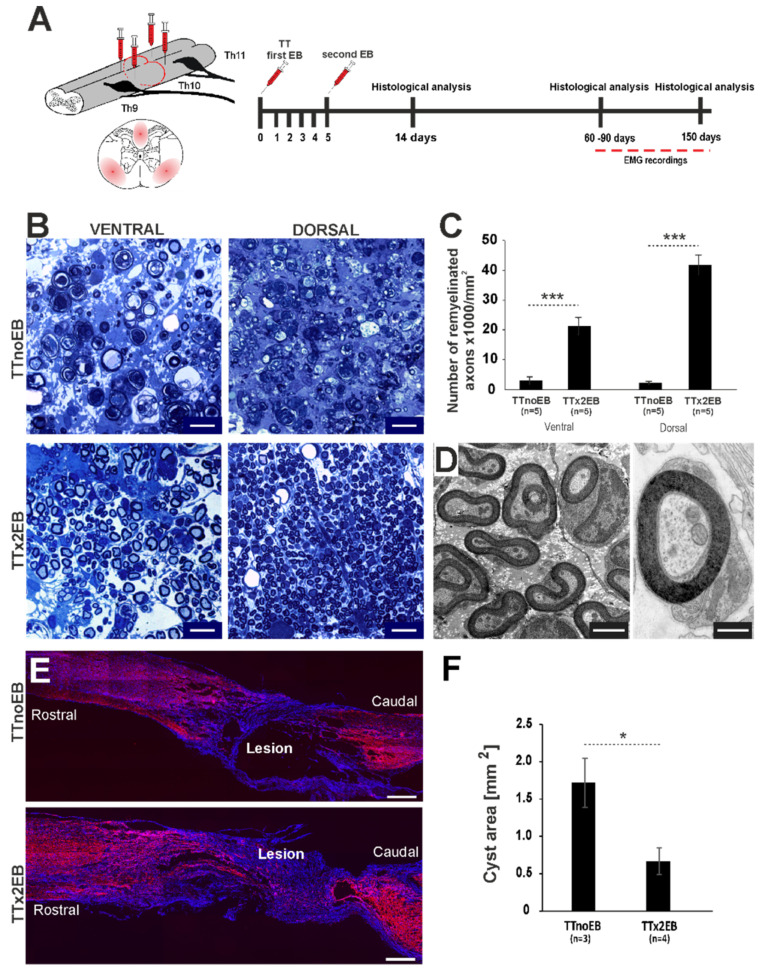
Experimentally induced white matter demyelination was followed by axon remyelination and regrowth through the transected spinal cord. (**A**) Schematic presentation of experimental design. Note the defined phases of experiments: single and double multifocal demyelination of dorsal and ventrolateral funiculi (first injection on the day of total transection TT and then second, 5 days later), histological examination at the early (14 dpi, days post-injury) and complete remyelination (60–90 dpi) and later at 150 dpi as well as functional examination with EMG recordings at 60–90 dpi. (**B**) Microscope images of semi-thin sections stained with toluidine blue showed efficient remyelination of white matter axons in the TTx2EB rats in contrast to limited remyelination observed in the control (TTnoEB) rats with total spinal cord transection but without any EB injections. Scale bars 10 μm. (**C**) Quantification revealed a significantly higher number of remyelinated axons in both dorsal and ventral white matter in double EB-injected spinal cords (TTx2EB, *n* = 5) compared to control ones (TTnoEB, *n* = 5). (**D**) Electron micrographs of ultrathin sections from remyelinated lesions showed myelin of peripheral identity and the presence of Schwann cells in the remyelinated spinal cord white matter caudally to the transection. Scale bars 1 μm (left image) and 500 nm (right image). (**E**) Representative longitudinal sections of spinal cords showing the injured tissue architecture (neurofilament in red, cell nuclei in blue) of rats from the control TTnoEB and TTx2EB groups. Scale bars 500 μm. (**F**) Significantly smaller cysts were present in the injured spinal cord from the TTx2EB rats (*n* = 4) in contrast to control TTnoEB rats (*n* = 3). Bar plots present means ± SEM. Statistical significance in comparison to control data from TTnoEB rats by Student’s *t*-test: * *p* < 0.05, *** *p* < 0.001.

**Figure 2 ijms-24-00495-f002:**
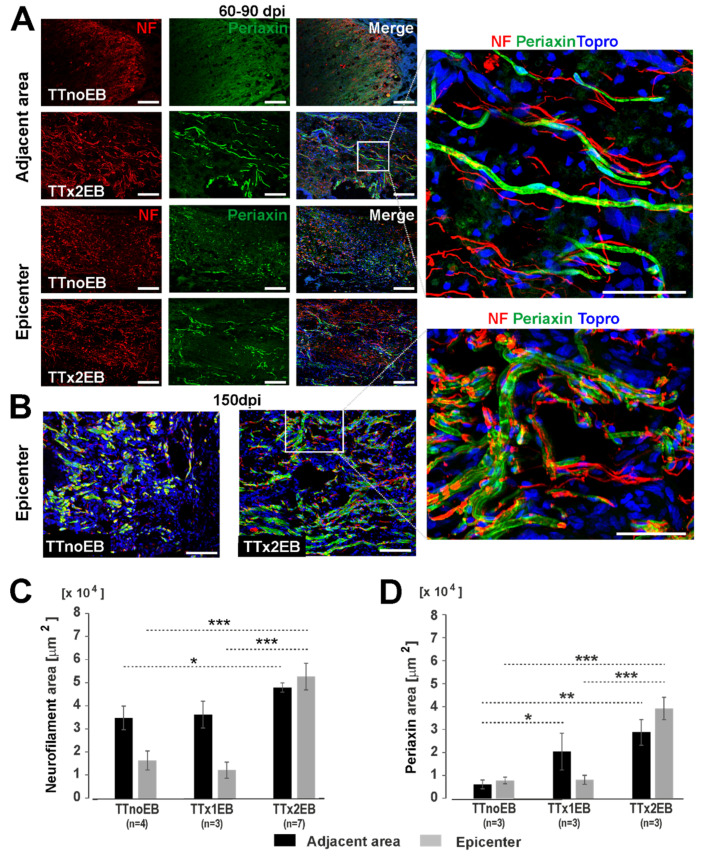
Representative images of axonal regeneration associated with Schwann-cell-derived remyelination through an injured spinal cord (**A**) at 60–90 days post-injury (dpi) and (**B**) 150 dpi (NF- neurofilaments in red, periaxin-Schwann-cell-specific myelin protein in green, cell nuclei in blue). Scale bars: 100 µm; higher magnification 50 µm. (**C**) Bar diagrams showing an increased area of neurofilament and (**D**) periaxin-positive staining at 60–90 dpi with demyelination induced by EB injection close to the total spinal cord transection. The bars present data as the mean ± SEM. The number of rats used in groups for comparison are indicated below the analyzed bars. Statistical significance of differences was checked with one-way ANOVA followed by post hoc Fisher’s LSD tests: * *p* < 0.05; ** *p* < 0.01, *** *p* < 0.001.

**Figure 3 ijms-24-00495-f003:**
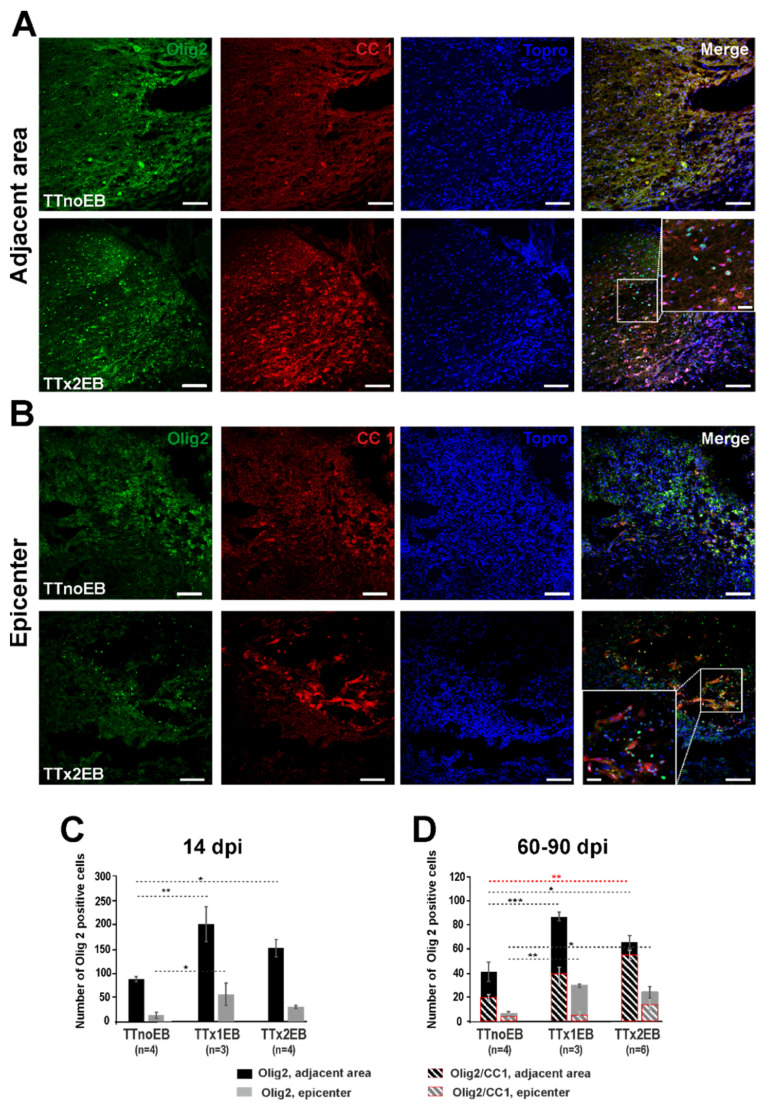
Multifocal demyelination induced by intraspinal injection of EB in close proximity to the spinal cord total transection site increased the number of oligodendrocyte lineage cells (Olig2) and mature oligodendrocytes (Olig2/CC1). Representative images of sections from spinal cord tissue adjacent to the injury (**A**) and at the injury epicenter (**B**) stained for Olig2^+^ (green), CC1^+^ (red), and cell nuclei (blue) at 60–90 days post-injury. Scale bars: low magnification—100 µm; higher magnification inlets—25 µm. Bar diagrams present a quantitative analysis of Olig2^+^ cells at 14 dpi (**C**) and Olig2^+^ and Olig2^+^/CC1^+^ cells (red outlines) at 60–90 dpi (**D**). The bars present data as the mean ± SEM. The number of rats used in groups for comparison are indicated below the analyzed bars. Statistical significance of differences was checked with one-way (**C**) and two-way (**D**) ANOVA followed by post hoc Fisher’s LSD tests: * *p* < 0.05; ** *p* < 0.01; *** *p* < 0.001 (in red, statistical significance of the Olig2^+^/CC1^+^ cell data analysis).

**Figure 4 ijms-24-00495-f004:**
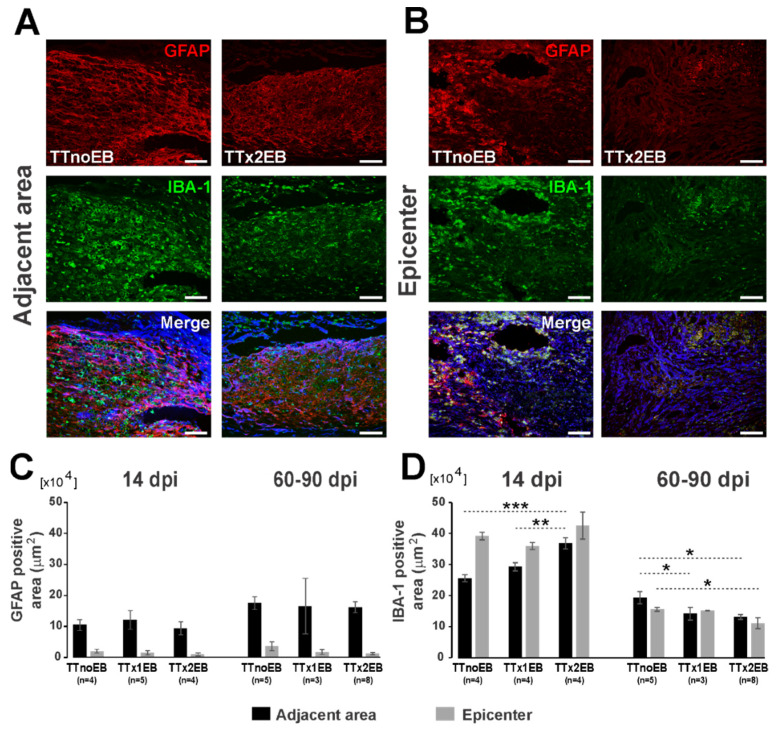
Local demyelination induced by intraspinal injection of EB in close proximity to the spinal cord total transection evoked limited changes of astroglia (GFAP) and microglia/macrophage (IBA-1) reactivity at 14 and 60–90 dpi (days post-injury). (**A**,**B**) Representative images of immunostaining for GFAP (red) and IBA-1 (green) in the spinal cord at injury-adjacent and epicenter areas at 60–90 dpi (cell nuclei in blue). Scale bars: 100 µm. (**C**,**D**) Bar diagrams show decrement of astroglia (GFAP) and microglia (IBA-1) at 14 and 60 dpi after demyelination. The bars present data as the mean ± SEM. The number of rats used in groups for comparison are indicated below the analyzed bars. Statistical significance of differences was checked for each type of staining and for each dpi period separately with one-way ANOVA followed by Fisher’s LSD post hoc tests: * *p* < 0.05, ** *p* < 0.01, *** *p* < 0.001.

**Figure 5 ijms-24-00495-f005:**
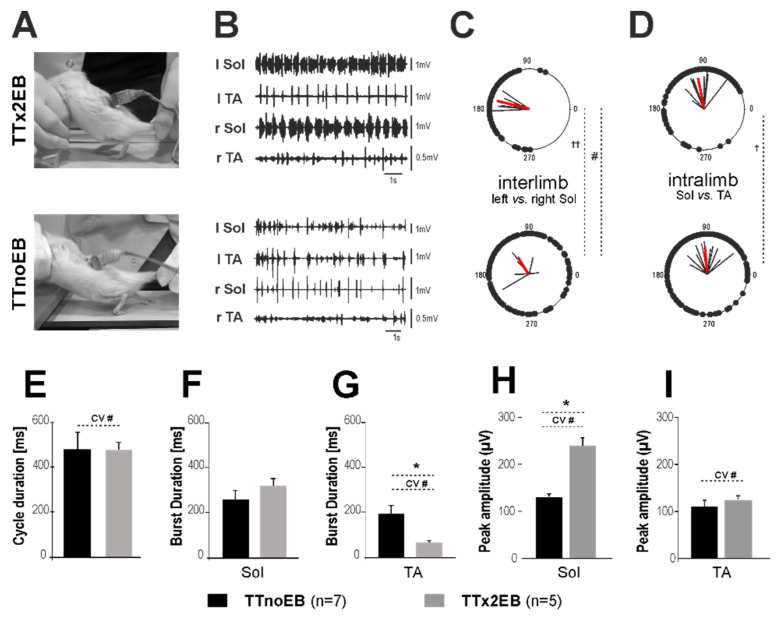
Improvement of locomotor performance in rats two to three months after chemically induced spinal cord remyelination by EB treatment after spinal cord total transection. (**A**) Representative video frames were taken during the locomotor performance on the treadmill of spinal rats from the TTx2EB group (**top** panel) with a typical locomotor posture induced by tail pinching and from the untreated TTnoEB group (**bottom** panel) with low posture and dragging limbs. (**B**) The accompanying EMG activity is presented on the (**top**,**bottom**) panels. (**C**,**D**) Polar plots of the interlimb and intralimb coordination established in spinal rats from the TTx2EB (**top**) and TTnoEB (**bottom**) groups. (**E**–**I**). Bar diagrams presenting the related results in the cycle duration (**E**), EMG burst duration of the Sol (**F**) and TA (**G**) muscles, and the EMG peak amplitude of the burst EMG activity of the Sol (**H**) and TA (**I**) muscles during hindlimb movements on the treadmill of spinal rats from the TTx2EB and TTnoEB groups. Statistical significance of ***r***-vector in polar plot analysis (**C**,**D**): for strength of coordination ^†^ *p* < 0.05 and ^††^ *p* < 0.01 (Mann-Whitney test); for phase shift differences ^#^ *p* < 0.05 Watson U2 test. The bars (E–I) present data as the mean ± SEM. Statistical significance in (E–I): * *p* < 0.05 (Student’s *t*-test), *n* = 5 (TTx2EB); *n* = 7 (TTnoEB). Statistical significance of the differences in the relative variability measured as a coefficient of variation (CV= SD/mean): CV # < 0.05 (Student’s *t*-test).

**Figure 6 ijms-24-00495-f006:**
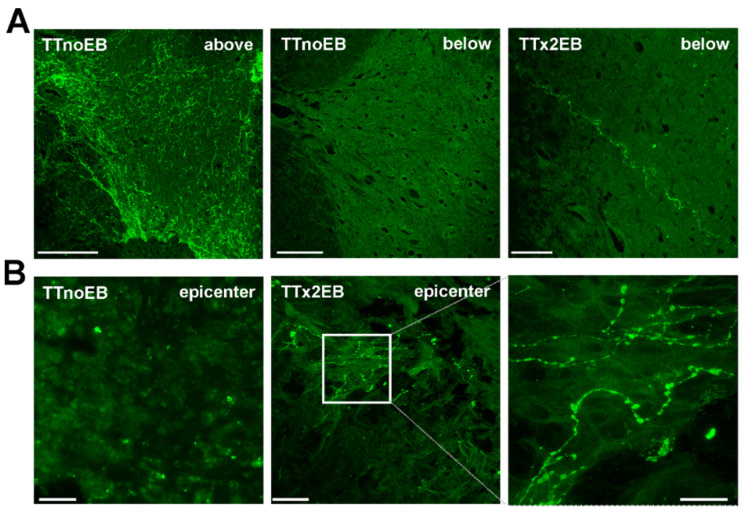
Representative images of serotonergic (5-HT^+^) innervation in the spinal cord cross-sections (**A**) above the transection site and below at the L5 level in the control (TTnoEB) and EB-treated rats (TTx2EB), as well as longitudinal sections (**B**) at the level of transection in the control (TTnoEB) and EB-treated rats (TTx2EB) at 60–90 dpi (days after injury). Scale bars: 100 µm (**A**); 20 µm (**B**).

**Figure 7 ijms-24-00495-f007:**
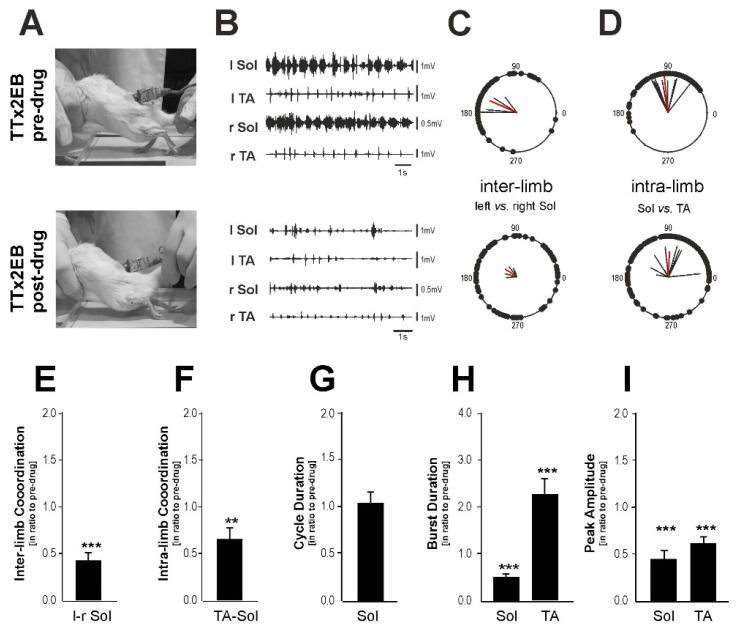
Alteration of hindlimb locomotor performance after blockade of 5-HT_2_ receptors in the TTx2EB rats by cyproheptadine application at 60–90 dpi (days after injury). (**A**) Representative frames of video recordings taken during the locomotor performance on the treadmill of the same spinal rat from the TTx2EB group in the pre-drug (**top** panel) and post-drug (**bottom** panel) conditions. (**B**) Examples of the corresponding EMG recordings in pre-drug (**top** panel) and post-drug (**bottom** panel) trials. (**C**,**D**) The interlimb ((**left** vs. (**right**) Sol) and intralimb (Sol vs. TA) coordination of hindlimb locomotor performance on the treadmill of spinal rats (*n* = 5) from the TTx2EB group in pre- (**top** panel) and post-drug (**bottom** panel) conditions. (**E**–**I**) Bar diagrams illustrating the effects of blockade of 5-HT_2_ receptors on strength of inter- (**E**) and intra-limb (**F**) coordination (***r***-value of circular analysis) as well as on cycle duration (**G**), EMG burst duration (**H**), and EMG peak amplitude (**I**) expressed in ratios to that of pre-drug ones. Data are presented as mean ± SEM, *n* = 5, statistical significance of differences by Student’s *t*-test: ** *p* < 0.005; *** *p* < 0.001.

## Data Availability

All data generated or analyzed during this study are included in this published article. Further inquiries can be directed to the corresponding authors.

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
