# Peer review of "Forced Remyelination Promotes Axon Regeneration in a Rat Model of Spinal Cord Injury"

_ijms, 2022, doi:10.3390/ijms24010495_

Round 1

Reviewer 1 Report

This is an excellent manuscript with an appropriate design backed up with relevant literature and a clear overall take home message. I only have a few comments that need to be considered in a revised version of the manuscript.

1. Concerning Fig1, why were the results of TT1xEB not shown here?

2. The authors should discuss more in depth why they think that chemically induced remyelination is more potent then the endogenous TT induced one; is it only about timing or are other pathways induced that could explain the difference?

3. in the discussion, I would like to read how these findings of forced remyelination would eventually be translated in the clinical setting. Would chemically induced demyelination be a treatment strategy also in humans or would the authors suggest to identify underlying pathways that will be therapeutically targeted? 

Author Response

ijms-2097930 - Forced remyelination promotes axon regeneration in a rat model of
spinal cord injury
by MaÅ‚gorzata Zawadzka*, Marine Yeghiazaryan, Sylwia NiedzióÅ‚ka,
Krzysztof Miazga, Anna Kwaśniewska, Marek Bekisz, Urszula Sławińska*

We are very pleased that the reviewers found our work interesting. We are very grateful for their insightful and helpful comments which we have addressed as follows.

This is an excellent manuscript with an appropriate design backed up with relevant literature and a clear overall take home message. I only have a few comments that need to be considered in a revised version of the manuscript.

  1. Concerning Fig1, why were the results of TT1xEB not shown here?

In our study, we first examined series of semi-thin sections of spinal cords from all experimental groups under the light and electron microscopy in order to assess the effectiveness of remyelination as well as tissue preservation in our experimental design. Results presented in Fig. 1 led us to propose that strong enhancement of OPC recruitment and differentiation which was possible to trigger by double toxin injection might trigger functionally-relevant effect. Therefore, in Fig.1 we intended to show the ultimate effects of such treatment on white matter remyelination and tissue preservation (described here as the significantly smaller areas of the lesion-created cysts). On the other hand, we have included all experimental groups in the analysis of every subsequent experiment that was designed to get insight into the mechanisms of described phenomenon. However, in most cases they showed any or negligible differences between controls (TT) and single-injected animals (TT1xEB). This further supports our hypothesis that strong enhancement of remyelination could have beneficial functional effect.

  1. The authors should discuss more in depth why they think that chemically induced remyelination is more potent then the endogenous TT induced one; is it only about timing or are other pathways induced that could explain the difference?

Our data proved that forced enhancement of endogenous remyelination could be beneficial for axonal protection/regeneration after spinal cord injury. In the light of our results this might be caused both by recruiting significantly more OPCs or/and Schwann cells into the lesion surrounding areas. Fig. 1 clearly shows that there is no spontaneous remyelination in TT cords in contrast to experimentally induced endogenous remyelination in TT2xEB animals.

  1. in the discussion, I would like to read how these findings of forced remyelination would eventually be translated in the clinical setting. Would chemically induced demyelination be a treatment strategy also in humans or would the authors suggest to identify underlying pathways that will be therapeutically targeted?

In our investigation we have mainly focused on describing the mechanisms in which experimentally forced remyelination of demyelinated white matter areas surrounding the transected area of the spinal cord supports axonal preservation/regeneration. We acknowledge that the question of the clinical relevance of the phenomenon we described remains unresolved. Undoubtedly, the experimental paradigm presented here needs to be overthought in order to be tested as a potential treatment strategy in the human clinic due to the toxicity of used chemical agent. However, we believe that our hypothesis of the translational potential of forced remyelination in spinal cord injury will be further extended and tested in more clinically-relevant experimental conditions. The novel results coming from the MS field seem to be very encouraging.

We have now discussed such possibilities in the revised manuscript as follows:

Recent advances in understanding the complex mechanisms of remyelination and the well-documented potential of its modulation achieved either by pharmacological promoting of endogenous OPCs migration and subsequent differentiation (https://pubmed.ncbi.nlm.nih.gov/26644513/, https://pubmed.ncbi.nlm.nih.gov/21131950/ ) or modulating the inflammatory reaction (https://pubmed.ncbi.nlm.nih.gov/36228613/, https://pubmed.ncbi.nlm.nih.gov/35822550/ )  could be also employed in spinal cord injury treatment.

Reviewer 2 Report

The paper describes a study of how forced demyelination caused by the administration of 2 doses of ethidium bromide (EB) promotes axon regeneration in a rat spinal cord injury model.

The article is a very thorough, well-executed and described study. The results are interesting and convincing.

I can mention the following problems. In the text and in several Figures, the quantitative analysis of the results of immunohistochemistry are shown for both options 1 and 2 injections of EB, but there are not indications about 1 injection on the slides themselves.

It would also be interesting to know why EB and this mode of its injections were chosen.

Were there any preliminary experiments?

If we look at the work in the perspective of using the results in clinical practice, what less toxic options are possible?

Author Response

ijms-2097930 - Forced remyelination promotes axon regeneration in a rat model of
spinal cord injury
by MaÅ‚gorzata Zawadzka*, Marine Yeghiazaryan, Sylwia NiedzióÅ‚ka,
Krzysztof Miazga, Anna Kwaśniewska, Marek Bekisz, Urszula Sławińska*

We are very pleased that the reviewers found our work interesting. We are very grateful for their insightful and helpful comments which we have addressed as follows.

The paper describes a study of how forced demyelination caused by the administration of 2 doses of ethidium bromide (EB) promotes axon regeneration in a rat spinal cord injury model.

The article is a very thorough, well-executed and described study. The results are interesting and convincing.

I can mention the following problems. In the text and in several Figures, the quantitative analysis of the results of immunohistochemistry are shown for both options 1 and 2 injections of EB, but there are not indications about 1 injection on the slides themselves.

In our study, we first examined series of semi-thin sections of spinal cords from all experimental groups under the light and electron microscopy in order to assess the effectiveness of remyelination as well as tissue preservation in our experimental design.  Results presented in Fig. 1 led us to propose that strong enhancement of OPC recruitment and differentiation which was possible to trigger by double toxin injection might trigger functionally-relevant effect. Therefore, in Fig. 1 we intended to show the ultimate effects of such treatment on white matter remyelination and tissue preservation (described here as the significantly smaller areas of the lesion-created cysts). On the other hand, we have included all experimental groups in the analysis of every subsequent experiment that was designed to get insight into the mechanisms of described phenomenon. However, in most cases they showed any or negligible differences between controls (TT) and single-injected animals (TT1xEB).  Therefore, we have provided data for all groups each time, but we did not include images where there was no statistical significance of differences. We decided to include only representative images clearly showing significant differences that support the results of the quantitative analysis.

It would also be interesting to know why EB and this mode of its injections were chosen.

Were there any preliminary experiments?

In our study, we took advantage of using a well-described toxin-based model of demyelination/remyelination. We injected transected cords with 1 ml of 0.1% EB into the dorsal and ventrolateral funiculi of the spinal cord to induce focal demyelination. Since the model has been extensively used and described in details by us (ex. Zawadzka et al., Cell Stem Cells, 2010) and many other authors and become one of the most frequently utilized rat model of demyelination/remyelination we did not present a proof of the accuracy of our experimental approach. Moreover, as we already mentioned in the result section of our submission, it has been described that when the same area is exposed to several rounds of demyelination/remyelination, the OPC numbers are not reduced and the efficiency of remyelination is not impaired by previous rounds of remyelination (Penderis et al., 2003). In our manuscript, we proved that two rounds of demyelination result in fact in enhancing remyelination and functional improvement.

If we look at the work in the perspective of using the results in clinical practice, what less toxic options are possible?

In our investigation, we have mainly focused on describing the mechanisms in which experimentally forced remyelination of demyelinated white matter areas surrounding the transected area of the spinal cord supports axonal preservation/regeneration. We acknowledge that the question of the clinical relevance of the phenomenon we described remains unresolved. Undoubtedly, the experimental paradigm presented here needs to be overthought in order to be tested as a potential treatment strategy in the human clinic due to the toxicity of the used chemical agent. However, we believe that our hypothesis of the translational potential of forced remyelination in spinal cord preservation/regeneration after injury will be further extended and tested in more clinically-relevant experimental conditions. The novel results coming from the MS field seem to be very encouraging.

We have now discussed such possibilities in the revised manuscript as follows:

Recent advances in understanding the complex mechanisms of remyelination and the well-documented potential of its modulation achieved either by pharmacological promoting of endogenous OPCs migration and subsequent differentiation (https://pubmed.ncbi.nlm.nih.gov/26644513/, https://pubmed.ncbi.nlm.nih.gov/21131950/ ) or modulating the inflammatory reaction (https://pubmed.ncbi.nlm.nih.gov/36228613/, https://pubmed.ncbi.nlm.nih.gov/35822550/) could be also employed in spinal cord injury treatment.